# Training load, sports performance, physical and mental health during the COVID-19 pandemic: A prospective cohort of Swiss elite athletes

Yannis Karrer[1]*, Stefan Fröhlich[2,3], Samuel Iff[1], Jörg Spörri[2,3], Johannes Scherr[2], Erich Seifritz[1], Boris B. Quednow[4], Malte Christian Claussen[1,5]

1 Department of Psychiatry, Psychotherapy and Psychosomatics, Psychiatric Hospital, University of Zurich, Zurich, Switzerland, 2 Department of Orthopaedics, Sports Medical Research Group, Balgrist University Hospital, University of Zurich, Zurich, Switzerland, 3 Department of Orthopaedics, University Centre for Prevention and Sports Medicine, Balgrist University Hospital, University of Zurich, Zurich, Switzerland, 4 Department of Psychiatry, Experimental and Clinical Pharmacopsychology, Psychotherapy and Psychosomatics, Psychiatric Hospital, University of Zurich, Zurich, Switzerland, 5 Private Clinic Wyss AG, Muenchenbuchsee, Switzerland

* yannis.karrer@yahoo.com

**Data Availability Statement:** All data files are available from Zenodo at https://doi.org/10.5281/zenodo.5655745.

## Abstract

### Background

The COVID-19 pandemic and associated restrictions have led to abrupt changes in the lives of elite athletes.

### Objectives

The objective of this prospective cohort study was to examine training load, subjective sports performance, physical and mental health among Swiss elite athletes during a 6-month follow-up period starting with the first Swiss lockdown.

### Methods

Swiss elite athletes (n = 203) participated in a repeated online survey evaluating health, training, and performance related metrics. After the first assessment during the first lockdown between April and May 2020, there were monthly follow-ups over 6 months.

### Results

Out of 203 athletes completing the first survey during the first lockdown, 73 athletes (36%) completed all assessments during the entire 6-month follow-up period. Sports performance and training load decreased during the first lockdown and increased again at the beginning of the second lockdown in October 2020, while symptoms of depression and financial fears showed only a transient increase during the first lockdown. Self-reported injuries and illnesses did not change significantly at any timepoint in the study. Stricter COVID-19 restrictions, as measured by the Government Stringency Index (GSI), were associated with

**Funding:** The authors received no specific funding for this work.

**Competing interests:** The authors have declared that no competing interests exist.

reduced subjective sports performance, as well as lower training intensity, increased financial fears, poorer coping with restrictions, and more depressive symptoms, as measured by the 9-item module of the Patient Health Questionnaire-9 (PHQ-9).

## Conclusion

This study revealed a negative impact of the COVID-19 restrictions on sports performance, training load and mental health among Swiss elite athletes, while the rate of self-reported injuries and illnesses remained unaffected.

## Introduction

The COVID-19 pandemic can be considered as one of the major challenges of our time. It led to numerous COVID-19 deaths and associated restrictions may have contributed to an increased loss of livelihoods. Moreover, the COVID-19 pandemic and associated government restrictions were concomitant with a growing mental health crisis in Europe [1, 2].

Mental health symptoms and disorders such as depression, anxiety, and sleeping disorders are common among elite athletes [3]. The COVID-19 pandemic has led to abrupt changes in the training routine and social support network that may be critical components for mental health problem coping in elite athletes [4, 5]. In addition, elite athletes may have been confronted with social isolation, postponement or cancellation of competitions, limited training possibilities, lack of meaning in daily activity, and fears concerning the continuation of an athletic career [6]. Therefore, elite athletes may represent a subpopulation especially challenged by the psychiatric burden of the COVID-19 pandemic [7].

Mental health impairments are capable of impairing performance, increasing the risk of musculoskeletal injuries, and prolonging the recovery from physical injuries [3]. Therefore, mental health is obviously strongly linked to physical health and sports performance.

The International Olympic Committee (IOC) consensus statement 2019 on mental health in elite athletes defines elite athletes as those athletes that compete at professional, Olympic or collegiate levels [3]. A recent systematic review found negative effects of the COVID-19 pandemic on overall physical fitness, training load, negative emotions and sleep quality among elite athletes [8]. However, elite athletes may be more resilient to the stressors of the COVID-19 pandemic compared with non-athletes [9] and novice athletes [10].

In Switzerland, the strictness of government restrictions [11] as well as the specific restrictions in professional sports varied substantially over time; professional sports were only allowed during the second lockdown [12, 13]. Therefore, a longitudinal study design over a longer period of time after the lockdown ended (including the beginning of the second lockdown) may add valuable information about the effects of the ongoing COVID-19 pandemic and associated restrictions.

Accordingly, the primary objective of this study was to examine sports performance among Swiss elite athlete during a 6-month follow-up starting at the beginning of the COVID-19 pandemic (first wave in Switzerland) and following until the beginning of the 2nd lockdown. Secondary objectives were the evaluation of mental health, physical health, training, and potential explanatory factors. We hypothesized that the lockdown has a negative impact on sports performance, mental health, physical health training and load among elite athletes, which is reversed upon lifting of the lockdown.

## Materials and methods

### Study design

This prospective cohort study was designed as a monthly 6-months follow-up during the COVID-19 pandemic in elite athletes. Participants entered their data online in an online questionnaire (REDCap 9.10.0 - © 2020 Vanderbilt University) [14]. The link to the follow-up questionnaires were each sent by email to the participants one month after the last survey was completed. If there was no response, a reminder email was sent 7 days after. The local ethics committee reviewed this study and judged it not to fall under the scope of the Swiss Human Research Act (HRA) by means of a declaration of non-responsibility (KEK-ZH-NR: Req-2020-00408). The data of this study were analysed anonymously, therefore no consent (written or oral) is needed.

### Study setting

For a detailed timeline of the study, refer to Fig 1. The first lockdown included the closure of sport facilities, public places, businesses, the restriction of contacts, the prohibition of any event, competition, or team training and asking the people to stay at home. These comprehensive actions resulted in the exposure of all participants to the "first lockdown". As part of the step-by-step reduction of restrictions, the closure of sport facilities ended on the 10th May 2020; team training was allowed again from 11th May 2020 [12].

The second lockdown included restrictions such as the restriction of contacts, the closure of public places, businesses as well as sport facilities on the 22nd of December 2020 [13]. In contrast to the first lockdown, professional athletes were still allowed to train in compliance with certain conditions and restrictions. There was no coherent definition of professional sports by the Swiss Government. As an objective measure to quantify the strictness of government policies regarding COVID-19 restrictions the government stringency index (GSI) was used [11].

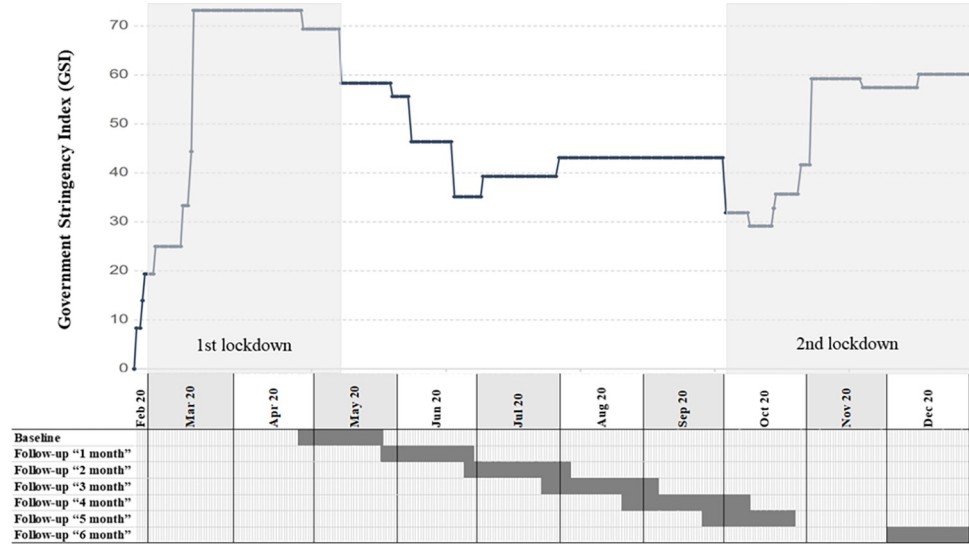

**Fig 1. Study timeline.** 1st lockdown: 1st March until 10th May 2020; 2nd lockdown: started on the 2nd October 2020 and did not end within the 6-month follow-up period of this study; First survey during lockdown: 25th April until 26th May 2020; Follow-up 1-month: 26th May until 29th June 2020; Follow-up 2-month: 26th June until 04th August 2020; Follow-up 3-month: 25th July until 6th September 2020; Follow-up 4-month: 24th August until 10th October 2020; Follow-up 5-month: 23rd September until 27th October 2020; Follow-up 6-month: 01st December 2020 until 04th January 2021. GSI [11].

The complete 6-month and partly the 5-month follow-up surveys were carried out during the 2nd lockdown which did not end within the 6-month follow-up period (FUP).

## Participants

A sample of adult elite athletes was recruited during the 25th of April 2020 and 25th of May 2020, of which baseline data were collected. An online questionnaire was distributed to elite level athletes of various sports via their respective sports clubs or national sports federation. In this study the same definition of elite athletes was used as defined in the 2019 IOC consensus statement on mental health in elite athletes [3]. A minimum training volume of 1 hour per day before the COVID-19 pandemic, a minimum age of 18 years and language skills in French or German were required to be included as a participant. Exclusion criteria were incomplete data, the participation in a non-Olympic sport or a sport that is not recognized by the IOC. The participants of this study did not receive any compensation for their participation.

## Demographic and sports variables

The collected variables consisted of gender, age, sports type, training volume in hours, subjective intensity, subjective athletic performance, existence of a secondary occupation inclusive workload during the lockdown, and then monthly until 6th month of follow-up. To evaluate status before lockdown, participants were asked to estimate following factors in the last month before the pandemic from memory: subjective performance (0–100% of subjective maximum performance), training volume (hours/day), training intensity (0–100% of subjective maximum intensity), financial fears were assessed by the question "Did you have financial fears?" and answered by a visual analogue scale (VAS) ranging from 0–100 (0 meaning "none"; 100 meaning "extreme"), alcohol (days/month) respectively cannabis consumption (days/month).

Additionally, the participants were asked if their income from sports is sufficient for their livelihood and independent of additional income (yes/no). Participants were also asked about own confirmed COVID-19 infections (yes/no). The subjective coping with the restrictions was assessed by the question "I am getting along well with the restrictions to limit the COVID-19 pandemic?" which was evaluated by VAS ranging from 0–100 (0 meaning "does not apply"; 100 meaning "does apply"). Worries for their sporting career was assessed by the question "Are your worrying about the continuation of your sporting career because of the COVID-19 pandemic?" which was evaluated by VAS ranging from 0–100 (0 meaning "no worries"; 100 meaning "extreme worries"). All questions about the demographic and sports variables were part of the survey during lockdown and all monthly surveys of the FUP.

## Clinical questionnaires

The validated German version of the short form (10 items) of the Spielberger State-Trait Anxiety Inventory (STAI) was used to obtain the anxiety rating [15]. The questionnaire investigates two types of anxiety: the state anxiety as a measure for the reaction to current events and the trait anxiety which represents hardly changeable personal characteristics. The 9-item module of the Patient Health Questionnaire (PHQ-9) as a reliable and valid tool was used to screen the participants for depressive symptoms [16]. As a validated tool to evaluate sleep disorder symptoms, the Insomnia Severity Index (ISI) was used to identify symptoms of actual sleep disturbances [17, 18]. The extracts from the Pittsburgh Sleep Quality Index served to obtain precise parameters of sleep (i.e. sleep latency, sleep duration) [19]. The Oslo Sports Trauma Research Center (OSTRC) questionnaire was used to assess current and already existing health problems like illness or injury [20–22]. All mentioned clinical questionnaires were part of the survey during lockdown and all monthly surveys of the FUP.

## Statistical methods

Parametric data was expressed as mean ± standard deviation (SD) and frequency tables for categorical data. Comparisons of demographic data between time points test was used for categorical tables with more than four fields, one-way analysis of variance for continuous variables, and Kruskal-Wallis rank test for variables nonmetric variables. Statistical significance was set at p <0.05. For sports performance, we fitted a general least squares random-effects model using the score of the time point as the dependent variable, pre-lockdown performance as the independent variable and gender, team sports, injury, activity, GSI, occupation percentage as covariates. To adjust for psychiatric measures, we added the results of PHQ-9, ISI, STAI, financial fears and worries for their sporting career. All explanatory variables that had an association with the independent variables at p<0.20 in the univariable analyses were included in the multivariable-adjusted analyses. Using a stepwise backward elimination process, the least significant variables were then removed from the base model. Only variables with p<0.05 remained in the final parsimonious model. Stata Statistical Software (Release 13, College Station, TX) was used to analyse the data.

## Patient and public involvement

There was no involvement of the public or the participants in the design, recruitment, conduct, choice of outcome measures, reporting or dissemination of this study. Participants that reported suicidal thoughts or at least moderate depressive symptoms were contacted by the study team; personal contact details were provided, and professional help was offered.

## Results

### Participants

During the first lockdown, 203 athletes completed the survey and were included in the analysis. Seventy-three (36%) athletes finalised the study, while 130 (64%) athletes were lost due to unknown reasons. The largest drop-out (37%) was seen between the first survey during the 1st lockdown and second survey 1-month after the 1st lockdown. After the initial drop-out, the monthly drop-out rate decreased until the end of the observation period and ranged between 3–18%. Descriptive data and characteristics of participating athletes at the first lockdown and the 6-month FUP are shown in Table 1. The age of participants ranged between 18–54 years (only two participants were >37 years old) at the first survey during lockdown and 18–37 years at all follow-up surveys. Overall, the age of participants, the ratio between male and female participants and the sufficiency of income did not change significantly over the observation period (at p<0.05).

**Table 1. Summary table of demographic data at different time points.**

|  | Lockdown | 1-month FUP | 2-month FUP | 3-month FUP | 4-month FUP | 5-month FUP | 6-month FUP |
|---|---|---|---|---|---|---|---|
|  | N = 203 | N = 127 | N = 107 | N = 88 | N = 80 | N = 75 | N = 73 |
| **Age at Survey Date** | 24.1 (± 5.3) years | 24.3 (± 5.0) years | 23.9 (± 4.6) years | 23.7 (± 4.5) years | 23.8 (± 4.5) years | 24.0 (± 4.4) years | 23.7 (± 4.5) years |
| **Female** | 45% | 50% | 53% | 47% | 48% | 51% | 48% |
| **Summer Sports** | 52% | 43% | 38% | 36% | 34% | 41% | 38% |
| **Winter Sports** | 48% | 57% | 62% | 64% | 66%) | 59% | 62% |
| **Working** | 60% | 59% | 61% | 61% | 59% | 63% | 62% |
| **Workload** | 67.1 (± 8.2) % | 66.0 (± 29.2) % | 64.8 (± 28.0) % | 65.0 (± 27.9) % | 61.7 (± 28.0) % | 65.6 (± 28.1) % | 64.0 (± 28.6) % |
| **Sufficient income** | 53% | 54% | 50% | 49% | 54% | 52% | 52% |

FUP: follow-up period.

### Primary analysis

**Sports performance.**   The subjective sports performance was significantly lower during the first lockdown and at the 1-month and 6-month follow-up compared to before the lockdown (Fig 2A). For the multivariable regression analysis of sports performance see Table 2.

### Secondary analysis

**Physical health and changes in training volume and intensity.**   At p<0.05, self-reported injury and illness did not change significantly over at any time point during this study. After the significant decrease of the training volume during the lockdown, training volume increased back to pre-lockdown levels (Fig 2B). An additional insignificant decrease (at p<0.05) in training volume was seen at the 6-month follow-up. Multivariable regression analysis ($R^2$ within = 0.0232, between = 0.04523, overall = 0.3003; Prob > $\chi^2$ = 0.0000) showed that the training volume was moderately positively associated with the training volume before the lockdown (B = 0.269, p<0.001), strongly positively associated with winter sport (vs. summer sport) (B = 0.499, p<0.001), and strongly negatively associated with occupation (B = -0.728, p<0.001), weakly negatively associated with age at survey date (B = -0.027, p<0.05) and the PHQ-9 sum (B = -0.077, p<0.001).

The subjective training intensity was significantly decreased during the lockdown and at the 1-month follow-up compared to pre-lockdown levels and decreased again significantly at the 6-month follow-up (Fig 2C). Multivariable regression analysis ($R^2$ within = 0.0953, between = 0.2114, overall = 0.1527; Prob > $\chi^2$ = 0.0000) revealed the training intensity to be weakly positively associated with training intensity before the lockdown (B = 0.225, p<0.001), weakly negatively associated with age at survey date (B = -0.601, p<0.01), strongly negatively

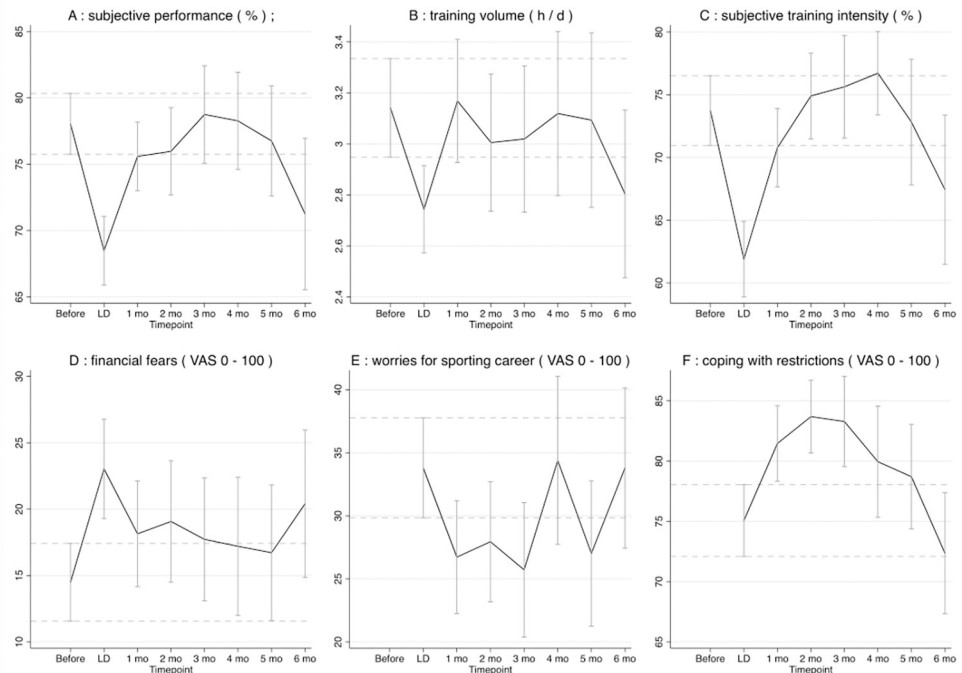

**Fig 2. Sports performance, training load, worries, fears, and coping.** (A) subjective performance (%); (B) training volume (h/d); (C) subjective training intensity (%); (D) financial fears (VAS 0–100); (E) worries for sporting career (VAS 0–100); (F) coping with restrictions (VAS 0–100). The dotted lines represent the 95% CI interval of the "before" measurement.

**Table 2. Multivariable and univariable regression analysis of subjective sports performance, financial fears, STATE sum and PHQ-9 sum.**

| Subjective performance (primary analysis) | | | State anxiety (STATE) sum (secondary analysis) | | |
| --- | --- | --- | --- | --- | --- |
| | Univariable | Multivariable | | Univariable | Multivariable |
| Self-reported injury/illness | -8.284*** | -7.337*** | Patient Health Questionnaire-9 (PHQ-9) sum | 2.067*** | 1.393*** |
| Patient Health Questionnaire-9 (PHQ-9) sum | -1.853*** | -1.512*** | Trait anxiety sum (TRAIT) | 0.705*** | 0.405*** |
| Government Stringency Index (GSI) | -0.258*** | -0.210*** | Insomnia Severity Sum Index (ISI) sum | 0.902*** | 0.320*** |
| Subjective performance before (%) | 0.218*** | 0.183** | Age at survey date | 0.078 | 0.157* |
| Insomnia Severity Sum Index (ISI) sum | -0.635*** | ns | Financial fears | 0.094*** | 0.031* |
| State anxiety (STATE) sum | -0.325*** | ns | Training volume (h/d) | -0.822*** | ns |
| Trait anxiety (TRAIT) sum | -0.267** | ns | Coping with restrictions | -0.065*** | ns |
| Financial fears | -0.099** | ns | Subjective training intensity (%) | -0.048*** | ns |
| | | | Government Stringency Index (GSI) | 0.048* | ns |
| $R^2$ within = 0.1100, between = 0.2475, overall = 0.1627; Prob > $\chi^2$ = 0.0000 | | | $R^2$ within = 0.2504, between = 0.7116, overall = 0.5958; Prob > $\chi^2$ = 0.0000 | | |
| Financial fears (secondary analysis) | | | Patient Health Questionnaire-9 (PHQ-9) sum (secondary analysis) | | |
| | Univariable | Multivariable | | Univariable | Multivariable |
| Occupation (vs. not) | -11.774*** | -6.452** | Insomnia Severity Sum Index (ISI) sum | 0.300*** | 0.164*** |
| Team sport (vs. individual) | -9.833** | -4.803* | State anxiety (STATE) sum | 0.165*** | 0.101*** |
| Patient Health Questionnaire-9 (PHQ-9) sum | 1.761*** | 0.959** | Trait anxiety (TRAIT) sum | 0.185*** | 0.077*** |
| Financial fears before | 0.753*** | 0.642*** | Subjective performance (%) | -0.033*** | -0.016*** |
| State anxiety (STATE) sum | 0.488*** | 0.218* | Coping with restrictions | -0.026*** | -0.010** |
| Government Stringency Index (GSI) | 0.150** | 0.121** | Government Stringency Index (GSI) | 0.025*** | 0.011* |
| Trait anxiety (TRAIT) sum | 0.601*** | ns | Financial fears | 0.027*** | 0.009* |
| Subjective performance (%) | -0.134*** | ns | Occupation (vs. not) | 1.013* | ns |
| Subjective training intensity (%) | -0.107*** | ns | Training volume (h/d) | -0.329*** | ns |
| Coping with restrictions | -0.106** | ns | Subjective training intensity (%) | -0.022*** | ns |
| $R^2$ within = 0.0624, between = 0.5971, overall = 0.4080; Prob > $\chi^2$ = 0.0000 | | | $R^2$ within = 0.3329, between = 0.6680, overall = 0.6063; Prob > $\chi^2$ = 0.0000 | | |

* $p < 0.05$

** $p < 0.01$

*** $p < 0.001$; ns: non-significant

associated with self-reported injury/illness (B = -5.219, p<0.01) and the Government Stringency Index (B = -0.311, p<0.001) and moderately negatively associated with the PHQ-9 sum (B = -1.374, p<0.001).

**Financial fears and worries about their sporting career.** After the significant increase of financial fears during the first lockdown compared to pre-lockdown levels, financial fears decreased and but increased again at the 6-month FUP (Fig 2D). At p<0.05, there was no significant difference between the financial fears during any time point of the FUP compared with before the lockdown, although financial fears were higher at every time point of the FUP. For the multivariable regression analysis of financial fears, see Table 2. The worries for their sporting career did not change significantly at any time point during this study (Fig 2E).

**Coping with government restrictions.** The subjective ability to cope with restrictions increased significantly during the first 3 month of the FUP but decreased again to lockdown baseline values (Fig 2F). At the end of the 6-month FUP, the ability to cope was even lower than during the first lockdown but did not reach statistical significance at p<0.05. Multivariable regression analysis ($R^2$ within = 0.0627, between = 0.2324, overall = 0.1213; Prob > $\chi^2$ = 0.0000) indicated that subjective coping with restrictions has a strong negative association with the PHQ-9 sum (B = -1.364, p<0.001) and a weak negative association with the GSI (B = -0.166, p<0.001) and worries for sporting career (B = -0.136, p<0.001).

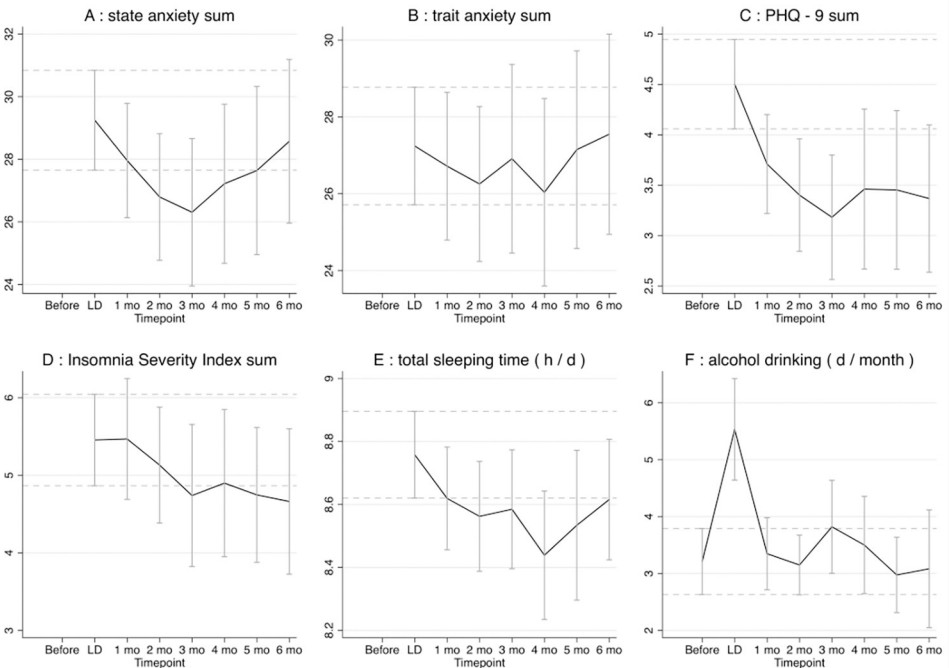

**Fig 3. Symptoms of anxiety, depression, sleep disturbances and alcohol consumption.** (A) state anxiety sum; (B) trait anxiety sum; (C) Patient Health Questionnaire-9 (PHQ-9) sum; (D) Insomnia Severity Index (ISI) sum; (E) total sleeping time (h/d); (F) alcohol drinking (d/month). The dotted lines represent the 95% confidence intervals of the first measurement of the variable, either during lockdown or from memory before lockdown.

The state anxiety (STATE) sum decreased significantly after the lockdown at the 2-month and 3-month time point before it increased again to baseline values of the lockdown (Fig 3A). For the multivariable regression analysis of the state anxiety sum, see Table 2. The trait anxiety (TRAIT) sum did not change significantly at any time point during this study (Fig 3B). With 4.5 (±3.1) points, the averaged PHQ-9 sum was highest during the first lockdown, interpreted as minimal symptoms of depression. There was a significant decrease at the 2-month and 3-month time point of the FUP compared to during the first lockdown, followed by a mild and insignificant increase of the PHQ-9 sum (Fig 3C). For the multivariable regression analysis of the PHQ-9 sum see Table 2. Neither the insomnia severity index (ISI) sum nor the total sleeping time (h) did change significantly at any time point of this study (Fig 3D and 3E).

**Alcohol and cannabis consumption.** Only two participants reported the consumption of cannabis of which the consumption frequency did not change at any time point during this study. Before the lockdown and in all follow-up surveys alcohol consumption was significantly lower than during the lockdown (Fig 3F). About half (range: 35–57%) of all participants reported not drinking any alcohol at each time point of this study.

**COVID-19 infections.** During the first lockdown until the 4-month survey of the FUP 1–2% of participants, respectively 4% at the 5-month and 9% at the 6-month survey reported own confirmed COVID-19 infections.

## Discussion

Among elite athletes, the negative effects on subjective sports performance, training load, symptoms of depression, financial fears, and alcohol consumption during the first lockdown returned back to pre-lockdown levels estimated from memory after the lockdown ended. At

the beginning of the 2nd lockdown the same factors changed, but only the reduction in sports performance and training intensity reached significance at p<0.05.

Our results regarding the period around the first lockdown seem to be in line with several previous studies that found reduced (objective) sports performance [23], reduced training load [24–28], negative impact on general mental state [24, 29, 30], increased negative emotions [25, 27, 31–33], increased risk of adjustment disorder [34] during lockdown among elite athletes. However, one study found increased sleeping time [28] and several previous studies found increased symptoms of insomnia during lockdown among elite athletes [25, 27, 28, 32, 33], which we both did not find in our study. We also did not find any gender specific increases in insomnia (or any other measures) like two previous studies [25, 33]. There was also a systematic review that found negative effect of the COVID-19 pandemic on overall physical fitness, training load, negative emotions and sleep quality among elite athletes [8]. In comparison to our study, that review did not include the beginning of a second lockdown and only examined the effects at the beginning of the pandemic.

Regarding the post-lockdown period, a study among elite rugby players did also find decreased depressive symptoms after the lockdown ended, but with no changes in symptoms of anxiety [35]. Because there was no comparison to pre-lockdown levels, it can only be speculated if symptoms of depression returned to pre-lockdown levels and if there was no effect on anxiety like in our study. Similar to our results, a prospective cohort study among Iranian elite athletes found decreased training load, positive mood and life satisfaction during lockdown compared to reopening phase and the following semi-lockdown [24]. They also found strongly increased economic damage (from 10.3% during lockdown in March 2020 to 52.3% in the reopening phase in May 2020 respectively 56.3% during the semi lockdown condition in July 2020) compared with pre-lockdown levels [24]. In contrast to our study, where training load returned to pre-lockdown levels, they found strongly decreased training load during all phases (-88% during lockdown, -85% during reopening, -86% during semi-lockdown) [24]. In Switzerland there were considerably stricter restrictions (as measured by the GSI) during the first lockdown [11]. Additionally, Iranian elite athletes only received privileges to exercise during the reopening phase [24]. In contrast, Swiss elite athletes received funds and the privileges to exercise shortly after the first lockdown ended [36], which likely reduced the negative impact of the COVID-19 restrictions on training load, financial fears. Although it is reasonable to think that financial fears and economic damage may correlate, we cannot exclude possible economic damage in Swiss elite athletes respectively changes in financial fears in Iranian elite athletes as they were not examined.

Our findings revealed a low number of participants that reported symptoms corresponding to moderate or moderately-to-severe depression, which was also found in one study among elite rugby players [35] but not in another study among professional football (i.e., soccer) players during the first lockdown [37]. Compared with rates of depression among elite athletes provided by the literature ranging from 4% [38] to 68% [39], the prevalence of increased PHQ-9 values in our cohort tends to the lower end of the range with 1–8% over the entire period of our study. In accordance with previous studies [9, 10], our study seems to support the hypothesis that a higher level of competition (i.e., elite level) may provide a protective effect on mental health and sports performance during the COVID-19 pandemic.

Although the reduced training load during the first lockdown may have offered an opportunity to cure existing injuries or illnesses, there were no significant changes in self-reported injuries or illnesses at any timepoint. One study also found no changes of the incidence of injuries among professional football (soccer) players comparing pre- with post-lockdown [40], whereas two other studies found lower injury rates in the 2020/21 season compared with normal seasons before the pandemic [41, 42]. In our study, self-reported injuries and illnesses

were unsurprisingly associated with lower subjective sports performance and lower training intensity. It can be assumed that sports performance and training intensity decrease in the presence of an injury or illness.

The investigation of potential explanatory factors for observed changes revealed several associations: stricter COVID-19 government restrictions reflected by a higher GSI were associated with lower subjective sports performance and training intensity, more depressive symptoms, increased financial fears and worse coping with restrictions. However, our study cannot answer if those associations reflect causation or correlation. More depressive symptoms measured by the PHQ-9 were found to be associated with worse subjective sports performance and coping with restrictions as well as more symptoms of anxiety (STATE, TRAIT), sleeping problems (ISI sum) and financial fears. Those findings seem plausible because mental health symptoms and athletic performance cannot be separated and anxiety and sleep problems are common symptoms of depression [3]. Besides an association of winter sports and training volume, no association between winter and summer sports with any tested parameter was found. The positive association between winter sports and training volume may be explained by the postponement of many competitions in summer sports. Also, the privileges to train again fell largely into the follow-up period that corresponds to the preparation for the winter sports season. Swiss elite athletes received funds and the privileges to exercise shortly after the first lockdown ended [36], which likely reduced the negative impact of the COVID-19 restrictions on sports performance, training and mental health.

## Methodological considerations

There are a couple of limitations one should consider when interpreting the study findings: first, as only questionnaires were used, the results therefore remain a subjective report of sports performance, training load and symptoms of mental and physical health. Clinical examination would be required for an accurate assessment of mental and physical health. Second, a certain degree of selection or reporting bias cannot be ruled out because mental health symptoms may increase or decrease the ability or willingness to participate. Third, the participants of this study represent a national cohort that was exposed to certain circumstances specific to elite athletes in Switzerland which may differ to other countries. Fourth, considering the collective of elite athletes representing only a small fraction of all athletes, the number of participants in this study was relatively large. The drop-out rate of about two thirds of participants over the 6-month FUP can be considered as high, however, not unusual for a follow-up study using online questionnaires only. However, drop-out participants were not significantly different from the participants that stayed in the study. Because we included every participant at any time point in the multivariable regression analysis, the effect of the high drop-out rate may be limited. We speculate that the main reasons for this high drop-out rate may be the losing of interest to participate because of the COVID-19 topic itself, the detailed survey or the relatively long FUP. Fifth, as with every questionnaire-based study we cannot rule out a potential recall bias. Sixth, the GSI only represents the restrictions overall and does not put extra weight on sport-specific restrictions [11] and the changes over time may have influenced the participants with unknown latency.

## Conclusion

This study revealed that the negative impacts of the COVID-19 pandemic and associated restrictions on the subjective sports performance, training load and mental health in Swiss elite athletes returned to pre-lockdown levels after the first lockdown ended, while no effects on self-reported injuries and illnesses were observed. A similar negative effect on training load

and sports performance was also observed after the beginning of the 2$^{nd}$ lockdown, but again with no changes of self-reported injuries and illnesses. Future research should also address if the observed effects on sports performance, training load, physical health, and mental health in the post-lockdown respectively the beginning of the second lockdown period also have occurred in other countries under similar or different conditions.

## Supporting information

**S1 Checklist. STROBE (Strengthening The Reporting of OBservational Studies in Epidemiology) checklist.**
(PDF)

## Acknowledgments

The participation of all the athletes was greatly appreciated.

## Author Contributions

**Conceptualization:** Stefan Fröhlich, Jörg Spörri, Johannes Scherr, Malte Christian Claussen.

**Data curation:** Stefan Fröhlich, Samuel Iff, Malte Christian Claussen.

**Formal analysis:** Yannis Karrer, Samuel Iff, Jörg Spörri, Johannes Scherr, Erich Seifritz, Boris B. Quednow, Malte Christian Claussen.

**Investigation:** Yannis Karrer, Stefan Fröhlich, Samuel Iff, Jörg Spörri, Johannes Scherr, Erich Seifritz, Boris B. Quednow, Malte Christian Claussen.

**Methodology:** Samuel Iff, Malte Christian Claussen.

**Project administration:** Yannis Karrer, Malte Christian Claussen.

**Software:** Samuel Iff.

**Supervision:** Boris B. Quednow.

**Visualization:** Yannis Karrer, Samuel Iff.

**Writing – original draft:** Yannis Karrer, Samuel Iff, Malte Christian Claussen.

**Writing – review & editing:** Yannis Karrer, Stefan Fröhlich, Samuel Iff, Jörg Spörri, Johannes Scherr, Erich Seifritz, Boris B. Quednow, Malte Christian Claussen.

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
