## [Decision Letter · Decision Letter 0]

21 Jul 2022

PONE-D-21-35814Sports performance, mental health, physical health, and training load during the progression of the COVID-19 pandemic: A cohort study of Swiss elite athletesPLOS ONE

Dear Dr. Claussen,

Thank you for submitting your manuscript to PLOS ONE. After careful consideration, we feel that it has merit but does not fully meet PLOS ONE’s publication criteria as it currently stands. Therefore, we invite you to submit a revised version of the manuscript that addresses the points raised during the review process.

We look forward to receiving your revised manuscript.

Kind regards,

Tauqeer Hussain Mallhi, Ph.D

Academic Editor

PLOS ONE

Journal Requirements:

2. Please ensure that you have specified (1) whether consent was informed, (2) what type you obtained (for instance, written or verbal, and if verbal, how it was documented and witnessed). If your study included minors, state whether you obtained consent from parents or guardians. If the need for consent was waived by the ethics committee and (3) If you are reporting a retrospective study of medical records or archived samples, please ensure that you have discussed whether all data were fully anonymized before you accessed them and/or whether the IRB or ethics committee waived the requirement for informed consent. If patients provided informed written consent to have data from their medical records used in research, please include this information.

3. PLOS requires an ORCID iD for the corresponding author in Editorial Manager on papers submitted after December 6th, 2016. Please ensure that you have an ORCID iD and that it is validated in Editorial Manager. To do this, go to ‘Update my Information’ (in the upper left-hand corner of the main menu), and click on the Fetch/Validate link next to the ORCID field. This will take you to the ORCID site and allow you to create a new iD or authenticate a pre-existing iD in Editorial Manager. Please see the following video for instructions on linking an ORCID iD to your Editorial Manager account: " ext-link-type="uri" xlink:type="simple">https://www.youtube.com/watch?v=_xcclfuvtxQ"

Reviewers' comments:

Reviewer's Responses to Questions

**Comments to the Author**

1. Is the manuscript technically sound, and do the data support the conclusions?

Reviewer #1: Yes

Reviewer #2: Yes

Reviewer #3: Yes

Reviewer #4: Partly

Reviewer #5: Yes

2. Has the statistical analysis been performed appropriately and rigorously? 

Reviewer #1: I Don't Know

Reviewer #2: Yes

Reviewer #3: Yes

Reviewer #4: Yes

Reviewer #5: Yes

3. Have the authors made all data underlying the findings in their manuscript fully available?

Reviewer #1: Yes

Reviewer #2: Yes

Reviewer #3: Yes

Reviewer #4: Yes

Reviewer #5: Yes

4. Is the manuscript presented in an intelligible fashion and written in standard English?

Reviewer #1: Yes

Reviewer #2: Yes

Reviewer #3: Yes

Reviewer #4: Yes

Reviewer #5: Yes

5. Review Comments to the Author

Reviewer #1: This study is well written with good methodological concept. Especially, the limitations section was well articulated.

I have some minor suggestions mentioned below-

1) Definition of elite athlete?

2) Line 82: "designed as a 6-month follow-up during the COVID-19 pandemic". Here, it should be highlighted that, this study actually did monthly follow up for 6 month.

3) Total number of participants should be mentioned at participants section

4) Dropout rate is very high, which is a major problem in this study. Though it was addressed and clarified in the limitations section, there is chance for further proper explanation regarding how the study is yet valid with this limitation. Hope, this will increase further the acceptance of this study.

5) Analysis part seemed to be authentic, but it is suggested for further comments from an expert reviewer

6) Line 273: Our findings revealed a low number of participants that reported symptoms indicating moderate to moderately-to-severe depression--- How it indicates depression level?

7) It would be better if conclusion section contains some new things or suggestion based on the study findings.

Reviewer #2: GENERAL COMMENTS

In this manuscript, the authors evaluated the relationship between COVID-19 pandemic and mental/physical health among the elite athletes. It was interesting that the value changed depending on the timing and the factors. However, the authors should clarify the definition of elite athlete in this study. Are they national team member? Although the authors revealed the differences of the prevalence between this study and other studies, it is difficult to evaluate about this.

Major

・P11 Line184-185, Figure 2a: Description about this part is insufficient. In this figure, there does not seem to have significant difference compared to before the lockdown. The authors need to show statistical numbers more specifically. And the authors need to show the meaning of dotted lines in the Fig. 2.

・P13 Line 201: How do you think about there is a difference between winter and summer sports?

・P15 Line 246: Suddenly the description about COVID-19 has begun. Please describe how to diagnose or assess the COVID-19 in the method.

Minor

P5 Line 109: The authors need to explain the abbreviation for FUP here

P14 Line 217-245: The authors need to indicate where these descriptions are related to the figure.

Reviewer #3: Is the scientific merit of the study high enough to justify publication?

No

Hypothesis:

No specific hypothesis is stated.

Purpose:

The objective of this prospective cohort study was to examine subjective sports performance, mental and physical health, training load among Swiss elite athlete during a 6-month follow-up period starting with the first Swiss lockdown in response to the COVID-19 pandemic.

Methods:

Swiss elite athletes (n=203) participated in a repeated online survey evaluating mental and physical health factors, as well as training and performance related metrics. After the first assessment during the first lockdown between April and May 2020, there were monthly follow-ups over a 6-month period as the pandemic progressed.

Are the methods appropriate for investigating the hypothesis?

There was no hypothesis specifically stated. Therefore, the methods do not look to investigate a particular hypothesis. However, the methods support the aim of the study.

Are the methods appropriate for investigating the stated goals?

Ideally they are. However, with only 36% of the initial cohort finishing the study, the effectiveness of the methodology (repeated online survey) must be evaluated.

Do the results support the hypothesis?

As no hypothesis was stated, the results cannot refute/accept a hypothesis.

Do the results support the purpose(s)?

No. With an appropriate response rate the findings of this paper could be evaluated vis-à-vis the remainder of the publications on the topic of the effect of COVID and its lockdowns on the mental and physical health of elite athletes. However, as this is grossly under-represented, the ability to draw conclusions based on the results in questionable.

Limitations:

1) Only questionnaires used, no clinical examinations

2) Possible selection bias of the limited number of respondents

3) Only Swiss elite athletes included

4) Only 36% of players completed the full study

5) Different sports may have had different restrictions, potentially greater than those mandated by the state lockdowns

Does the study make a large enough contribution to the existing literature to justify publication?

It does not right now. There have been a number of papers on the same basic topic with the same result which have a higher academic veracity due to adequate responses and/or clinical data. There is little in this paper which would affect my decision-making or standard of care.

Is the study suitable for publication in PLOS ONE?

No, for reasons stated above.

Reviewer #4: Dear Authors,

Kindly address the following questions and make the amendments accordingly:

1. Why there are two (2) titles? Suggest to change the title to “Training load, sports performance, physical and mental health during the COVID-19 pandemic: A prospective cohort of Swiss elite athletes.

2. The abstract’s format is wrong: supposed to be written in one paragraph and begin with the introduction or background of the study.

3. In sentence 30, grammatical error: …… among Swiss elite “athletes”……

4. In sentence 38, suggest to change to “Out of 203 athletes…….”

5. In sentence 53, …… can be considered as one of the major challenges……

6. In sentence 58, there were too many references (3 – 6) for a well-known short fact, suggest to limit to 1 or 2 reference(s).

7. In sentence 62, again, too many references were cited (7 – 11) for such a short sentence, suggest to limit to 1 or 2 reference(s).

8. In sentence 69, grammatical mistake: and an increased “in”……

9. The whole introduction only focused on the literature reviews of the negative impact and contributing factors of mental health in elite athletes, how about the effect of the COVID-19 pandemic on physical health, training load and sports performance among elite athletes from other studies?

10. In sentence 84, the link to the follow-up questionnaires was sent “by” email…….

11. The local ethics committee judged the study did not fall under the scope of the Swiss Human Research Act, so is there any ethical approval by any institutional review board or independent ethics committee before conducting the study? Do the participants of the study give written consent?

12. In sentence 90, refer to “Figure 1”.

13. In sentences 94 and 95, kindly standardize in writing the date: 10th May 2020; 11th May 2020

14. In sentence 97, kindly standardize in using a capital letter, i.e. “lockdown” instead of Lockdown

15. In sentences 99 – 102, why there was variation in terms of duration of follow-up from 2nd follow-up onward, unlike monthly follow-up as you have mentioned? Also, why is the word “GSI” appear at the end of the sentence 102?

16. In sentence 105, …… professional sports were still allowed to train in compliance with……

17. In sentence 109, replace FUP with follow-up.

18. In sentence 111, a sample of adult elite athletes “aged between ? and ?” was recruited from 25th April 2020 till 25th May 2020…….

19. In sentence 113, ......via their respective sports club or national sports federation.

20. Kindly revise and be more specific in the inclusion criteria: “a minimum training volume of 1 hour per day before the COVID-19 pandemic”. For example, if an elite athlete only trained for ½ hour one day before the lockdown, do you recruit this athlete?

21. Why do you exclude elite athletes from non-IOC-recognized sports? Are they non considered elite athletes even if they are representing the country to compete and participate in international championship? (selection bias)

22. How do you calculate your sample size? What is the sampling method?

23. In sentence 124, 0 – 100% of subjective “measurement”, not maximum

24. In sentence 125, existential fears were assessed as only having financial fear, is this reliable and validated? Please cite the reference that you used in assessing the existential fears.

25. In sentence 127, (……100 meaning “extreme fear”), alcohol (days/ month) and cannabis (days/ month) consumption. Why you did not include cigarette smoking?

26. In sentence 132, ……which was evaluated by VAS……

27. In sentence 133, worries for their "sporting" career……

28. In sentence 137, the FUP abbreviation supposes to be mentioned earlier before using it in the sentence 109.

29. In sentences 153 and 154, ……and Kruskal-Wallis test is non-parametric test for not normally distributed variables.

30. In sentence 162, Stata Statistical Software (Release 13, College Station, TX).

31. In sentence 177, what do you mean by “the rate of occupation”, aren’t all the elite athletes working as full-time athletes?

32. For tablet 1, suggest to divide into Male and Female for each column (each follow-up period)

33. In sentence 185, remove the word “mark”, and spell “Figure” in a complete word

34. In sentence 195, remove the word “mark” (please do proper proofreading)

35. In sentence 235, grammatical mistake: time point

36. The discussion can be improved and written clearer. For example, the initial paragraphs mainly discussed own findings/ results, and only the last two paragraphs compared with other studies/ literature.

37. In sentence 289, is it selection bias or reporting bias?

38. Can this study represent all the elite national athletes as there was selection bias in terms of sports from the beginning? Kindly address this as one of the limitations, and also mention another limitation is recall bias in a questionnaire-based study.

39. Do you attempt to improve the retention and reduce the dropout rate since you already foresee a high degree of dropout for a questionnaire-based study? Kindly discuss how can you improve the retention?

Thank you for your cooperation.

Reviewer #5: In the submitted paper the authors investigate the subjective sports

performance, mental and physical health, training load among Swiss elite athlete during a 6-month

follow-up period starting with the first Swiss lockdown in response to the COVID-19 pandemic in a prospective cohort study.

The study revealed a negative impact of the COVID-19 restrictions on sports performance,

training load and mental health among Swiss elite athletes, while the rate of self-reported injuries and illnesses remained unaffected.

Methods and statistics of the study are adequate and state of the art.

However, the results of this study are not surprising and are in line with other studies analyzing the impact of Lock Downs in the general public. Moreover, the paper is descriptive and I am missing some recommendations for future look downs. Also it is specific to the swiss situation which was a quite mild lockdown situation compared to other countries, . So, it may be interesting to compare these data sets with data sets from other countries with more stringent lockdowns. So, all in all data which can be published but the question is whether the significance of the data I such high to justify that it is published in Plos one.

6. PLOS authors have the option to publish the peer review history of their article (what does this mean?). If published, this will include your full peer review and any attached files.

Reviewer #1: **Yes: **Dr. Panchanan Acharjee

Reviewer #2: No

Reviewer #3: No

Reviewer #4: **Yes: **Alston Choong

Reviewer #5: No

---

## [Author Response · Author response to Decision Letter 0]

20 Sep 2022

Reviewer #1: This study is well written with good methodological concept. Especially, the limitations section was well articulated.

Thank you very much!

I have some minor suggestions mentioned below-

Ad 1) Definition of elite athlete?

We used the same definition of elite athletes as defined in the IOC consensus statement on mental health in elite athletes (Reardon CL, Hainline B, Aron CM, et al. Br J Sports Med 2019;53:667–699.).

To prevent misconceptions we added this definition in the Introduction “The International Olympic Committee (IOC) consensus statement 2019 on mental health in elite athletes defines elite athletes as those athletes that compete at professional, Olympic or collegiate levels (3).” and clarified that this definition was used in our study in the Methods: “In this study the same definition of elite athletes was used as defined in the 2019 IOC consensus statement on mental health in elite athletes (3).”

Ad 2) Line 82: "designed as a 6-month follow-up during the COVID-19 pandemic". Here, it should be highlighted that, this study actually did monthly follow up for 6 month.

Thank you, we corrected this sentence.

Ad 3) Total number of participants should be mentioned at participants section

We strictly adhere to the rule to report numbers in the results section, not in the methods section (see STROBE guidelines). Total number of participants is described in the first paragraph of the results. 

Ad 4) Dropout rate is very high, which is a major problem in this study. Though it was addressed and clarified in the limitations section, there is chance for further proper explanation regarding how the study is yet valid with this limitation. Hope, this will increase further the acceptance of this study.

Despite best efforts to keep the respondent number high during follow up, the 6th month follow-up included only roughly 1/3 of the original participants. This is inherent in the longitudinal design of the study. We reported all dropout rates accordingly. Despite selective follow-up participation, there was minimal indication of selection bias, data were non-significant among participants, dropouts, and the entire cohort. Hence, we think that the overall conclusion remains valid from this cohort.

Ad 5) Analysis part seemed to be authentic, but it is suggested for further comments from an expert reviewer

We leave this decision at the discretion of the editor.

Ad 6) Line 273: Our findings revealed a low number of participants that reported symptoms indicating moderate to moderately-to-severe depression--- How it indicates depression level?

Thank you for your comment. We changed the wording to “corresponding to”.

The PHQ-9 as an instrument to detect depression symptoms has been validated in elite athletes (Gouttebarge V, et al. International Olympic Committee (IOC) Sport Mental Health Assessment Tool 1 (SMHAT-1) and Sport Mental Health Recognition Tool 1 (SMHRT-1): towards better support of athletes' mental health. Br J Sports Med. 2021 Jan;55(1):30-37. doi: 10.1136/bjsports-2020-102411. Epub 2020 Sep 18. PMID: 32948518; PMCID: PMC7788187). 

Following cut-offs are widely used: ≥5=mild, ≥10=moderate, ≥15=moderately severe and ≥20=severe (Manea, L.; Gilbody, S.; McMillan, D. Optimal cut-off score for diagnosing depression with the Patient Health Questionnaire (PHQ-9): A meta-analysis. Can. Med. Assoc. J. 2012, 184, E191–E196).

Ad 7) It would be better if conclusion section contains some new things or suggestion based on the study findings.

We added following suggestion for future research: “Future research should also address if the observed effects on sports performance, training load, physical health, and mental health in the post-lockdown respectively the beginning of the second lockdown period also have occurred in other countries under similar or different conditions.”

 

Reviewer #2: In this manuscript, the authors evaluated the relationship between COVID-19 pandemic and mental/physical health among the elite athletes. It was interesting that the value changed depending on the timing and the factors. However, the authors should clarify the definition of elite athlete in this study. Are they national team member? Although the authors revealed the differences of the prevalence between this study and other studies, it is difficult to evaluate about this.

We used the same definition of elite athletes as defined in the IOC consensus statement on mental health in elite athletes (Reardon CL, Hainline B, Aron CM, et al. Br J Sports Med 2019;53:667–699.).

To prevent misconceptions we added this definition in the Introduction “The International Olympic Committee (IOC) consensus statement 2019 on mental health in elite athletes defines elite athletes as those athletes that compete at professional, Olympic or collegiate levels (3).” and clarified that this definition was used in our study in the Methods: “In this study the same definition of elite athletes was used as defined in the 2019 IOC consensus statement on mental health in elite athletes (3).”

Major

・P11 Line184-185, Figure 2a: Description about this part is insufficient. In this figure, there does not seem to have significant difference compared to before the lockdown. The authors need to show statistical numbers more specifically. And the authors need to show the meaning of dotted lines in the Fig. 2.

The dotted lines represent the 95% CI interval of the “before” measurement, so that readers can spot differences from the baseline more easily. So, subfigures A, B, C , D and F were significantly different from baseline values. We are happy to provide a table with the results in the supplementary material, however we think the current graph allows readers to grasp the development over time more easily. 

・P13 Line 201: How do you think about there is a difference between winter and summer sports?

Thank you for this input. We added a brief discussion about the potential explanations: “The positive association between winter sports and training volume may be explained by the postponement of many competitions in summer sports. Also, the privileges to train again fell largely into the follow-up period that corresponds to the preparation for the winter sports season.”

・P15 Line 246: Suddenly the description about COVID-19 has begun. Please describe how to diagnose or assess the COVID-19 in the method.

The participants were asked about own confirmed COVID-19 infections (yes/no question). COVID-19 infections are usually confirmed by rt-PCR. We added following sentence to the methods section “Participants were also asked about own confirmed COVID-19 infections (yes/no).” 

Minor

P5 Line 109: The authors need to explain the abbreviation for FUP here

Thank you, we explained the abbreviation in the mentioned sentence.

P14 Line 217-245: The authors need to indicate where these descriptions are related to the figure.

We added the related figures where they were missing in the results.

Reviewer #3: Is the scientific merit of the study high enough to justify publication?

No

Hypothesis:

No specific hypothesis is stated.

Our hypothesis was that the lockdown has a negative impact on sports performance, mental health, physical health training and load among elite athletes, which is reversed upon lifting of the lockdown.

For clarification we added following sentence in the introduction: “We hypothesized that the lockdown has a negative impact on sports performance, mental health, physical health training and load among elite athletes, which is reversed upon lifting of the lockdown.”

Purpose:

The objective of this prospective cohort study was to examine subjective sports performance, mental and physical health, training load among Swiss elite athlete during a 6-month follow-up period starting with the first Swiss lockdown in response to the COVID-19 pandemic.

Methods:

Swiss elite athletes (n=203) participated in a repeated online survey evaluating mental and physical health factors, as well as training and performance related metrics. After the first assessment during the first lockdown between April and May 2020, there were monthly follow-ups over a 6-month period as the pandemic progressed.

Are the methods appropriate for investigating the hypothesis?

There was no hypothesis specifically stated. Therefore, the methods do not look to investigate a particular hypothesis. However, the methods support the aim of the study.

Our hypothesis was that the lockdown has a negative impact on sports performance, mental health, physical health training and load among elite athletes, which is reversed upon lifting of the lockdown.

For clarification we added following sentence in the introduction: “We hypothesized that the lockdown has a negative impact on sports performance, mental health, physical health training and load among elite athletes, which is reversed upon lifting of the lockdown.”

Are the methods appropriate for investigating the stated goals?

Ideally they are. However, with only 36% of the initial cohort finishing the study, the effectiveness of the methodology (repeated online survey) must be evaluated. 

Drop-out participants were not significantly different from the participants that stayed in the follow-up. Therefore, we added following sentence to the limitations: “However, drop-out participants were not significantly different from the participants that stayed in the study.”

“Because we included every participant at any time point in the multivariable regression analysis, the effect of the high drop-out rate may be limited.”

Do the results support the hypothesis?

As no hypothesis was stated, the results cannot refute/accept a hypothesis.

Do the results support the purpose(s)?

No. With an appropriate response rate the findings of this paper could be evaluated vis-à-vis the remainder of the publications on the topic of the effect of COVID and its lockdowns on the mental and physical health of elite athletes. However, as this is grossly under-represented, the ability to draw conclusions based on the results in questionable.

Because it is always important to consider all relevant limitations when drawing conclusions from any study, we addressed all relevant limitations (including the drop-out rate). Drop-out participants were not significantly different from the participants that stayed in the follow-up. We therefore think that the ability to draw conclusions based on the results of our study is there.

Limitations:

1) Only questionnaires used, no clinical examinations

2) Possible selection bias of the limited number of respondents

3) Only Swiss elite athletes included

4) Only 36% of players completed the full study

5) Different sports may have had different restrictions, potentially greater than those mandated by the state lockdowns

Ad 1) Questionnaires were all validated and are used in clinical context. We are unsure which clinical examinations would have added to the primary objective of sports performance among Swiss elite athletes. We understand that individual tests per athlete could have given a more detailed information about performance, but those tests are specific for the sport and cannot be compared with each other.

Ad 2) This limitation is discussed in the limitation section. We are aware that there can be a selection bias. However, we still think that the results add to the literature in a meaningful way.

Ad3) Correct.

Ad 4) Correct. However, the data was missing at random, and we have no indication that the results are biased by those loss to follow up.

Ad 5) The different effects on the different sports were taken into account by evaluation of the training time and intensity.

Does the study make a large enough contribution to the existing literature to justify publication?

It does not right now. There have been a number of papers on the same basic topic with the same result which have a higher academic veracity due to adequate responses and/or clinical data. There is little in this paper which would affect my decision-making or standard of care.

Is the study suitable for publication in PLOS ONE?

No, for reasons stated above.

Thank you for your comments.

We want to highlight that we submitted this manuscript almost 1 year ago in November 2021 and waited since then for the completion of the review process respectively a first decision. Since several studies were published during this waiting period, we performed an updated literature search. We included the relevant publications that we found by the updated literature search in our revised manuscript.

Although there have been a number of papers on the same basic topic, we could not find any other prospective cohort study that examined the beginning of a second lockdown and/or studies that observed participants over a similar or longer period of time after the second lockdown ended. We also could not find any prospective cohort study that examined sports performance, training load, physical and mental health among elite athletes including all of the following time periods: pre- lockdown (1st lockdown), during lockdown (1st lockdown), post lockdown (first lockdown), beginning of the second lockdown. We therefore think that our study adds to the body of evidence in a meaningful way that justifies a publication in PLOS ONE.

We also think that it is valuable to have data from different countries with different situations regarding restrictions, funds/privileges to exercise for elite athletes. With more studies from different countries, patterns may emerge that may affect future decision-making of physicians as well as politicians and standard care.

Reviewer #4: Dear Authors,

Kindly address the following questions and make the amendments accordingly:

1. Why there are two (2) titles? Suggest to change the title to “Training load, sports performance, physical and mental health during the COVID-19 pandemic: A prospective cohort of Swiss elite athletes.

Thank you for this valuable input. We changed the title and removed the second subtitle.

2. The abstract’s format is wrong: supposed to be written in one paragraph and begin with the introduction or background of the study.

We corrected the format and added a study background part.

3. In sentence 30, grammatical error: …… among Swiss elite “athletes”……

Corrected.

4. In sentence 38, suggest to change to “Out of 203 athletes…….”

Corrected.

5. In sentence 53, …… can be considered as one of the major challenges……

Corrected.

6. In sentence 58, there were too many references (3 – 6) for a well-known short fact, suggest to limit to 1 or 2 reference(s).

We limited the references to one reference as suggested.

7. In sentence 62, again, too many references were cited (7 – 11) for such a short sentence, suggest to limit to 1 or 2 reference(s).

We limited the references to two references as suggested.

8. In sentence 69, grammatical mistake: and an increased “in”……

Corrected.

9. The whole introduction only focused on the literature reviews of the negative impact and contributing factors of mental health in elite athletes, how about the effect of the COVID-19 pandemic on physical health, training load and sports performance among elite athletes from other studies?

Thank you for this valuable input.

We complemented the introduction and discussion with up-to-date literature on mental health as well as training load, physical health and sports performance.

10. In sentence 84, the link to the follow-up questionnaires was sent “by” email…….

Corrected.

11. The local ethics committee judged the study did not fall under the scope of the Swiss Human Research Act, so is there any ethical approval by any institutional review board or independent ethics committee before conducting the study? Do the participants of the study give written consent?

The local Ethics Committee is an independent ethics committee. Therefore, no additional ethical approval is necessary according to Swiss Law. The data of this study were analysed anonymously, therefore no consent (written or oral) is needed.

12. In sentence 90, refer to “Figure 1”.

According to our understanding of the PLOS ONE Manuscript Body Formatting Guidelines (https://journals.plos.org/plosone/s/submission-guidelines#loc-figures-and-tables), abbreviations like “Fig” oder “Figs” should be used instead of Figure/Figures.

13. In sentences 94 and 95, kindly standardize in writing the date: 10th May 2020; 11th May 2020

Corrected.

14. In sentence 97, kindly standardize in using a capital letter, i.e. “lockdown” instead of Lockdown

Corrected.

15. In sentences 99 – 102, why there was variation in terms of duration of follow-up from 2nd follow-up onward, unlike monthly follow-up as you have mentioned? Also, why is the word “GSI” appear at the end of the sentence 102?

The variation in terms of duration of follow-up are explained by individually different dates of the first survey completion. The first survey during lockdown was completed between 25th April until 26th May 2020. If a participant completed the first survey on 1st May 2020, he would receive the 1-month follow-up survey by mail on 29th May 2020 (1 month after the first survey). Depending on how fast the participant responded to each survey, the timeframes may vary a bit. 

As a part of figure 1, the GSI (“government stringency index”) appears in the legend of figure 1 to cite the source.

16. In sentence 105, …… professional sports were still allowed to train in compliance with……

Changed to “[…] professional athletes were still allowed to train in compliance with […]”.

17. In sentence 109, replace FUP with follow-up.

Changed to “follow-up period”.

18. In sentence 111, a sample of adult elite athletes “aged between ? and ?” was recruited from 25th April 2020 till 25th May 2020…….

We added the report of age ranges at each survey in the “Participants” section of the results: “The age of participants ranged between 18-54 years (only two participants were 37 years old) at the first survey during lockdown and 18-37 years at all follow-up surveys.”

19. In sentence 113, ......via their respective sports club or national sports federation.

Corrected.

20. Kindly revise and be more specific in the inclusion criteria: “a minimum training volume of 1 hour per day before the COVID-19 pandemic”. For example, if an elite athlete only trained for ½ hour one day before the lockdown, do you recruit this athlete?

If an athlete only trained ½ hour per day before the pandemic, he would be excluded according to our inclusion criteria. 

21. Why do you exclude elite athletes from non-IOC-recognized sports? Are they non considered elite athletes even if they are representing the country to compete and participate in international championship? (selection bias)

We changed the wording from “the participation in a non-Olympic sport or a sport that is not recognized by the IOC.” to “a sport that is not recognized by the IOC.” as there was no athlete that competes in a IOC recognized but non-Olympic sport and we aimed to examine elite athletes competing in IOC recognized sports.

We used the same definition of elite athletes as defined in the IOC consensus statement on mental health in elite athletes (Reardon CL, Hainline B, Aron CM, et al. Br J Sports Med 2019;53:667–699.).

To prevent misconceptions we added this definition in the Introduction “The International Olympic Committee (IOC) consensus statement 2019 on mental health in elite athletes defines elite athletes as those athletes that compete at professional, Olympic or collegiate levels (3).” and clarified that this definition was used in our study in the Methods: “In this study the same definition of elite athletes was used as defined in the 2019 IOC consensus statement on mental health in elite athletes (3).”

22. How do you calculate your sample size? What is the sampling method?

There was no sample size calculation possible as no such event like a pandemic lockdown was ever present before and we couldn’t estimate the magnitude of the effect on the outcome variable. So, we used a convenience sample in this study.

23. In sentence 124, 0 – 100% of subjective “measurement”, not maximum

0-100% subjective maximum refers to “0-100% of subjective maximum performance” respectively “0-100% of subjective maximum intensity”. 

24. In sentence 125, existential fears were assessed as only having financial fear, is this reliable and validated? Please cite the reference that you used in assessing the existential fears.

It is correct that we aimed to examine existential fears in the sense of financial fears but explicitly asked for financial fears. To prevent misconceptions or accidentally using invalid methods, we changed existential fears to financial fears in the whole paper. 

25. In sentence 127, (……100 meaning “extreme fear”), alcohol (days/ month) and cannabis (days/ month) consumption. Why you did not include cigarette smoking?

We agree that tobacco or nicotine containing products (such as cigarettes) could have also been asked in the survey. We did not include smoking, because from our clinical experience elite athletes may indeed use tobacco (in particular in the form of spit tobacco/snus) or other nicotine containing products, but rarely smoke cigarettes. On the other hand, alcohol and cannabis consumption in elite athletes has been reported relatively often. Additionally, we speculate that the potential risk of recall bias may be higher when asking about the monthly use of tobacco/nicotine containing products compared to alcohol and cannabis because tobacco/nicotine containing products are more often consumed multiple times daily.

26. In sentence 132, ……which was evaluated by VAS……

Corrected.

27. In sentence 133, worries for their "sporting" career……

Corrected.

28. In sentence 137, the FUP abbreviation supposes to be mentioned earlier before using it in the sentence 109.

Corrected.

29. In sentences 153 and 154, ……and Kruskal-Wallis test is non-parametric test for not normally distributed variables.

That’s right. In case a non-normal distributed variable was present, for non-parametric variables, this test was used to analyse the data.

30. In sentence 162, Stata Statistical Software (Release 13, College Station, TX).

Changed.

31. In sentence 177, what do you mean by “the rate of occupation”, aren’t all the elite athletes working as full-time athletes?

Not all elite athletes have sufficient income from sports for their livelihood and therefore have an additional occupation. See table 1 for percentages of athletes with sufficient income from sports for each timepoint.

To clarify and prevent misconceptions, we changed the sentence 177 from “the rate of occupation and the sufficiency of income did not change significantly over the observation period (at p0.05).” to “the sufficiency of income did not change significantly over the observation period (at p0.05).”

32. For tablet 1, suggest to divide into Male and Female for each column (each follow-up period)

There were no significant differences by gender in any of the evaluated variables, hence we did not analyse the genders separately. 

33. In sentence 185, remove the word “mark”, and spell “Figure” in a complete word

We removed every “mark” in the manuscript.

According to our understanding of the PLOS ONE Manuscript Body Formatting Guidelines (https://journals.plos.org/plosone/s/submission-guidelines#loc-figures-and-tables), abbreviations like “Fig” or “Figs” should be used instead of Figures.

34. In sentence 195, remove the word “mark” (please do proper proofreading)

We removed every “mark” in the manuscript.

35. In sentence 235, grammatical mistake: time point

Changed.

36. The discussion can be improved and written clearer. For example, the initial paragraphs mainly discussed own findings/ results, and only the last two paragraphs compared with other studies/ literature.

Thank you for this valuable input.

We want to highlight that we submitted this manuscript almost 1 year ago in November 2021 and waited since then for the completion of the review process respectively a first decision. Since several studies were published during this waiting period, we performed an updated literature search. We included the relevant publications that we found by the updated literature search in our revised manuscript. Furthermore, we improved the discussion section by discussing our results in the context of other studies and overall comprehensive revision of the discussion section.

37. In sentence 289, is it selection bias or reporting bias?

Both biases cannot be ruled out. Therefore, we added “or reporting bias”. 

38. Can this study represent all the elite national athletes as there was selection bias in terms of sports from the beginning? 

They way of recruiting did not ensure a representative sample of all elite national athletes. However, drop-out participants were not significantly different from the participants that stayed in the follow-up. Therefore, we added following sentence to the limitations “However, drop-out participants were not significantly different from the participants that stayed in the study.” Potential selection bias is already addressed in the limitations section.

Kindly address this as one of the limitations, and also mention another limitation is recall bias in a questionnaire-based study.

We also added this limitation in the limitations section: “Fifth, as with every questionnaire-based study we cannot rule out a potential recall bias.”

39. Do you attempt to improve the retention and reduce the dropout rate since you already foresee a high degree of dropout for a questionnaire-based study? Kindly discuss how can you improve the retention?

We attempted to improve retention by a reminder mail. If the participants did not respond to the questionnaire sent by email, they received a reminder email 7 days after (see section “Study Design” 5-6th line). Our study was designed on short term notice as soon as the lockdown was foreseeable, and the study protocol did not foresee any additional measures to improve retention.

Thank you for your cooperation.

Thank you for your valuable inputs!

Reviewer #5: In the submitted paper the authors investigate the subjective sports performance, mental and physical health, training load among Swiss elite athlete during a 6-month follow-up period starting with the first Swiss lockdown in response to the COVID-19 pandemic in a prospective cohort study.

The study revealed a negative impact of the COVID-19 restrictions on sports performance, training load and mental health among Swiss elite athletes, while the rate of self-reported injuries and illnesses remained unaffected.

Methods and statistics of the study are adequate and state of the art.

However, the results of this study are not surprising and are in line with other studies analysing the impact of Lock Downs in the general public. Moreover, the paper is descriptive and I am missing some recommendations for future look downs. Also it is specific to the swiss situation which was a quite mild lockdown situation compared to other countries, . So, it may be interesting to compare these data sets with data sets from other countries with more stringent lockdowns. So, all in all data which can be published but the question is whether the significance of the data I such high to justify that it is published in Plos one.

Thank you very much for your time and valuable input.

Most but not all of our results are in line with other studies examining the first lockdown among elite athletes. However, to not publish studies that are not surprising or contradicting existing evidence leads to publication bias which we aim to prevent.

Because there has never been a comparable situation, it is important to monitor and describe the effects of such an extraordinary situation on several variables among elite athletes over time. 

We speculated that for example receiving funds and privileges to exercise may have reduced the impact of the COVID-19 restrictions on sports performance, training, and mental health. However, our study cannot answer if there was a causal relationship which would justify to make clear recommendations regarding COVID-19 related restrictions regarding elite athletes. 

As you stated correctly, our study is specific to the Swiss situation which you have considered as a quite mild lockdown situation comparing to other countries. The valuation of a “mild” lockdown is highly depending on the compared country. As an example: Switzerland had a GSI (as an objective measure to quantify the strictness of government policies regarding the COVID-19 restrictions) of about 73/100 vs. China with about 81/100 vs. Iran with about 59/100 during the first lockdown. Because the restrictions varied so much between countries/regions and over time, we think that our study provides valuable data for future comparisons with longitudinal data of other countries with similar or different restrictions. Even though we also think it would be interesting to compare our data with data sets of other countries, such a comparison was out of the scope of our study.

Although, there have been a number of papers on the same basic topic, we could not find any other prospective cohort study in elite athletes on the same topic that examined the beginning of a second lockdown and/or studies that observed participants over a similar or longer period of time after the second lockdown ended. We also could not find any prospective cohort study that examined sports performance, training load, physical and mental health among elite athletes including all of the following time periods: pre- lockdown (1st lockdown), during lockdown (1st lockdown), post lockdown (first lockdown), beginning of the second lockdown. We therefore think that our study adds to the body of evidence in a meaningful way that justifies a publication in PLOS ONE.

---

## [Decision Letter · Decision Letter 1]

14 Nov 2022

Training load, sports performance, physical and mental health during the COVID-19 pandemic: A prospective cohort of Swiss elite athletes

PONE-D-21-35814R1

Dear Dr. Karrer,

We’re pleased to inform you that your manuscript has been judged scientifically suitable for publication and will be formally accepted for publication once it meets all outstanding technical requirements.

Kind regards,

Tauqeer Hussain Mallhi, Ph.D

Academic Editor

PLOS ONE

Additional Editor Comments (optional):

Thank you for revising the manuscript.

Reviewers' comments:

Reviewer's Responses to Questions

**Comments to the Author**

1. If the authors have adequately addressed your comments raised in a previous round of review and you feel that this manuscript is now acceptable for publication, you may indicate that here to bypass the “Comments to the Author” section, enter your conflict of interest statement in the “Confidential to Editor” section, and submit your "Accept" recommendation.

Reviewer #2: All comments have been addressed

Reviewer #4: All comments have been addressed

2. Is the manuscript technically sound, and do the data support the conclusions?

Reviewer #2: Yes

Reviewer #4: Yes

3. Has the statistical analysis been performed appropriately and rigorously? 

Reviewer #2: Yes

Reviewer #4: Yes

4. Have the authors made all data underlying the findings in their manuscript fully available?

Reviewer #2: Yes

Reviewer #4: Yes

5. Is the manuscript presented in an intelligible fashion and written in standard English?

Reviewer #2: Yes

Reviewer #4: Yes

6. Review Comments to the Author

Reviewer #2: (No Response)

Reviewer #4: Well done to the authors as all my queries have been explained and addressed well. Although the research topic is not something novelty as there were already many publications with the similar topic, however the outcomes and results from a Swiss elite athletes (can be considered as a niche population) may be able to provide some additional information especially when the number of COVID-19 infection cases are still fluctuating with the latest emergent of Omicron subvariant XBB and BQ1. Thus, more original studies will be able to provide insights, suggestions, recommendations and served as a references to develop more up-to-date, state-of-art guideline in case there is a further lockdown due to another wave of COVID-19 subvariant pandemic or any other infectious disease pandemic.

7. PLOS authors have the option to publish the peer review history of their article (what does this mean?). If published, this will include your full peer review and any attached files.

Reviewer #2: No

Reviewer #4: **Yes: **Alston Choong

quillbot-extension-portal/quillbot-extension-portal

---

## [Editor Report · Acceptance letter]

21 Nov 2022

PONE-D-21-35814R1 

Training load, sports performance, physical and mental health during the COVID-19 pandemic: A prospective cohort of Swiss elite athletes 

Dear Dr. Karrer:

I'm pleased to inform you that your manuscript has been deemed suitable for publication in PLOS ONE. Congratulations! Your manuscript is now with our production department. 

Kind regards, 

on behalf of

Dr. Tauqeer Hussain Mallhi 

Academic Editor

PLOS ONE